# The Tissue Architecture of Oral Squamous Cell Carcinoma Visualized by Staining Patterns of Wheat Germ Agglutinin and Structural Proteins Using Confocal Microscopy

**DOI:** 10.3390/cells10092466

**Published:** 2021-09-18

**Authors:** Estefania Silveyra, Ronell Bologna-Molina, Rogelio Gónzalez-Gónzalez, Miguel Arocena

**Affiliations:** 1Molecular Pathology, School of Dentistry Universidad de la República (UDELAR), Las Heras 1925, Montevideo 14600, Uruguay; esilveyra@iibce.edu.uy; 2Department of Research, School of Dentistry, Universidad Juárez del Estado de Durango, Durango 34070, Mexico; rogelio.gonzalez@ujed.mx; 3Biochemistry and Biophysics, School of Dentistry Universidad de la República (UDELAR), Las Heras 1925, Montevideo 14600, Uruguay; 4Genomics Department, Instituto de Investigaciones Biológicas Clemente Estable (IIBCE), Montevideo 11600, Uruguay

**Keywords:** oral squamous cell carcinoma, wheat germ agglutinin, tyramide signal amplification, tissue architecture

## Abstract

Objectives: Tissue architecture and cell morphology suffer profound alterations during oral cancer and are important markers for its progression and outcome. For precise visualization of tissue architecture in oral cancer, we used confocal microscopy to examine the staining pattern of wheat germ agglutinin, a lectin that binds membrane glycoproteins, and the staining patterns of structural proteins. Materials and Methods: Paraffin sections of oral squamous cell carcinoma were stained with fluorescently labeled wheat germ agglutinin and with antibodies against structural proteins, which were revealed by immunohistochemistry with tyramide signal amplification. Results: Membrane localization of wheat germ agglutinin was markedly decreased in the basal layers and in regions of tumor invasion, accompanied by cytoplasmic redistribution of E-cadherin, β-actin and syndecan-1. Wheat germ agglutinin staining clearly identified tumor clusters within the surrounding stroma, and tumor cells with elongated morphology. Conclusions: Our results suggest that the wheat germ agglutinin staining pattern is indicative of the degree of cell cohesion in oral squamous cell carcinoma, which decreases in basal layers and invasive tumor clusters with more migratory morphologies. Wheat germ agglutinin staining in combination with confocal microscopy could constitute, therefore, a valuable tool for the study of tissue architecture in oral cancer.

## 1. Introduction

Tumor growth and its progressive taking over of neighboring tissues impose drastic alterations in tissue architecture, which have prognostic value in many types of cancers [1,2], including oral squamous cell carcinoma (OSCC). Invasion patterns in OSCC, for instance, can be classified as a wide pushing border, infiltrating cell islets, thin infiltrating cords or individual infiltrating cells [3], with the last two patterns being associated with worse prognoses [4]. Therefore, refining our understanding of OSCC tissue architecture is likely to have an important clinical impact.

High resolution microscopy, such as confocal microscopy, can help to overcome the limitations of conventional bright-field microscopy in the analysis of tumor tissue structure [5], and studies which explore its uses in the context of OSCC can make significant contributions towards understanding the pathobiology of this tumor [6]. In addition to confocal microscopy, the use of markers that permit visualizing tissue architecture clearly will be an important feature of studies that seek to obtain detailed structural descriptions of OSCC at the tissue level. Wheat germ agglutinin (WGA) is a plant lectin with high affinity for N-acetylglucosamine and sialic acid [7], which can bind to multiple membrane glycoproteins containing these monosaccharides in their varying carbohydrate moieties, and in particular it can bind to cell adhesion molecules [8,9,10]. Fluorescently labeled WGA has been used as a powerful stain for tissue architecture in combination with confocal microscopy [11,12]. In particular, altered tissue architecture, especially at the level of cell–cell adhesion, produces a clearly altered pattern of WGA staining [11]. Another important consideration for studies of tumor architecture with high resolution microscopy is the frequently mentioned difficulty of performing immunofluorescence staining in formalin-fixed, paraffin embedded (FFPE) thin sections [13], which is relevant given the prevalence of archival FFPE tumor samples, but this can be circumvented, for example, by the use of signal amplification techniques such as tyramide signal amplification (TSA), which generates covalently bound fluorophores in very close proximity to the detected epitope [14].

In this study, we have used confocal microscopy, in combination with fluorescently labeled WGA and immunohistochemistry with TSA, for fluorescent staining of structural proteins, to visualize tissue architecture in FFPE thin sections obtained from archival OSCC tumor samples. We have observed alterations in the staining patterns of WGA and structural proteins consistent with loss of cell cohesion in the basal layers of the oral mucosa, and in regions of tumor projections towards the underlying stroma, accompanied by changes in cell morphology suggestive of a migratory phenotype. Additionally, WGA staining readily allows to distinguish clusters of invasive tumor cells from the surrounding stroma. Our results therefore show the usefulness of WGA staining for the study of OSCC tissue architecture which, in combination with immunohistochemistry accompanied by TSA for fluorescent staining of structural proteins, and the use of confocal microscopy, permits clear visualization of the changes in tissue cohesion and cell morphology as tumor invasion progresses. 

## 2. Materials and Methods

### 2.1. Tissue Samples

FFPE tissue samples of normal oral mucosa (*n* = 3) and OSCC (*n* = 5) from the archives of the Molecular Pathology area of the School of Dentistry, Universidad de la República (Uruguay) were used.

### 2.2. Immunohistochemistry and Confocal Microscopy

After deparaffinization, 5 μm sections were heat retrieved with Reveal Decloaker solution (Biocare Medical, Pacheco, CA, USA) and endogenous peroxidases were blocked with 0.9% hydrogen peroxide for 5 min. Sections were incubated with either of the following primary antibodies in 1:100 dilution: pan-cytokeratin (clone AE1/AE3, Biocare Medical), Ki-67 antibody (clone MIB-1, DAKO, Carpinteria, CA, USA), E-cadherin antibody (clone EP6, Biocare Medical), β-actin antibody (clone 13E5, Cell Signaling, Danvers, MA, USA), syndecan-1 (CD138) antibody (clone B-A38, Bio SB, Santa Barbara, CA, USA) and vimentin antibody (clone V9, Biocare Medical). After primary antibody incubation, sections were incubated with a biotinylated secondary antibody followed by a streptavidin–horseradish peroxidase complex (Mouse/Rabbit ImmunoDetector Biotin Link, Bio SB, Santa Barbara, CA, USA). Next, sections were incubated for 10 min with CY3 or fluorescein labeled tyramide (TSA Plus Cyanine 3 Kit or TSA Plus Fluorescein Kit, Perkin Elmer, Waltham, MA, USA), washed with phosphate buffered saline (PBS) and incubated with either CF^®^488A conjugated WGA or CF^®^640R conjugated WGA (Biotium, Fremont, CA, USA) at 1:400 dilution for 20 min. Alternatively, 5 μm sections can be stained directly with fluorescently labeled WGA without the previous immunohistochemistry steps [12]. For double antibody labeling, the first primary-secondary antibody complex was removed by heat retrieval and the procedure was repeated for the second primary antibody. In some slides, nuclei were counterstained by RedDotTM Far-Red Nuclear Stain (Biotium). Sections were visualized with a Zeiss LSM 800 confocal microscope, and images were acquired with 20× and 40× objectives. 

## 3. Results

Fluorescently labeled WGA staining of normal oral mucosa reveals a highly ordered pattern of cell layers, with cell boundaries strongly stained and sharply demarcated (Figure 1A), including the Ki-67 positive cells of the basal layer (Figure 1A, inset). In OSCC, clearly demarcated cell boundaries can be observed in the more external layers of the oral mucosa by WGA staining (Figure 1B), but the staining becomes more cytoplasmic and diffuse in the basal layers, where Ki-67 positive cells are found (Figure 1B, inset), and in particular, cell boundaries are not easily distinguished.

In regions of tumor invasion into the stroma, tumoral cell clusters of varying size could be clearly distinguished from the surrounding stroma by pan-cytokeratin staining (Figure 2A,C). Simultaneous WGA staining marked both tumor clusters and the surrounding stromal compartment, but it also allowed us to clearly visualize epithelial cell shapes within the tumor clusters, thereby distinguishing them from the adjacent stromal cells (Figure 2B). In some tumor clusters, the demarcation of cell boundaries by WGA staining was particularly sharp (Figure 2D). Interestingly, tumor clusters could be seen to co-express pan-cytokeratin and vimentin (Figure 3A,B), and large vimentin positive cells with epithelial shape in areas of tumor invasion can also be seen to co-express Ki-67 (Figure 3C,D).

Next, we co-stained normal oral mucosa and OSCC mucosa with WGA and structural proteins. In normal oral mucosa, the proteins E-cadherin, β-actin and syndecan-1 display similar staining patterns to that of WGA, clearly marking cell boundaries (Figure 4A,C,E, respectively; see also Appendix A). In OSCC mucosa, however, the diffuse WGA staining observed in the basal layers is accompanied by marked increases in cytoplasmic E-cadherin, β-actin and syndecan-1, and also by a blurring of cell boundaries not seen in normal oral mucosa (Figure 4B,D,F, respectively, and insets in each figure). Moreover, in regions of tumor invasion, tumoral cell clusters with diffuse WGA staining at cell boundaries and large amounts of cytoplasmic E-cadherin, β-actin and syndecan-1 can be observed (Figure 5), a pattern that was not observed in normal oral mucosa at the same magnification (Appendix A). Interestingly, tumor cells with large amounts of cytoplasmic E-cadherin, β-actin and syndecan-1 frequently switch from a polygonal cell shape to a highly elongated shape, suggestive of increased migratory capacity, and this elongated morphology can also be seen against the surrounding stroma by WGA staining (arrows in Figure 5).

Our observations of WGA staining patterns in OSCC show that loss of WGA staining at cell boundaries is frequently accompanied by E-cadherin, β-actin and syndecan-1 relocalization from the cell periphery to the cytoplasm. E-cadherin and β-actin play fundamental roles in cell–cell adhesion, and syndecan-1 is highly involved in this process as well. Decreased expression of these proteins at cell boundaries in the basal layers of OSCC mucosa and in regions of tumor invasion suggests that loss of cell cohesion occurs in these tumor areas. The simultaneous transition from a WGA staining pattern marking cell boundaries sharply to a more diffuse staining pattern could also, thereby, constitute an indication of loss of cell cohesion.

## 4. Discussion

In this study, we used fluorescently labeled WGA in combination with TSA staining of structural proteins and confocal microscopy to visualize tissue architecture in FFPE thin sections of OSCC. WGA staining sharply marks the cell boundaries in the more external layers, whereas it becomes diffuse in the basal layers, paralleling a switch from periphery to cytoplasm in the distribution of the structural proteins E-cadherin, β-actin and syndecan-1. WGA staining allows to distinguish clearly between tumor cell clusters and the surrounding stroma in regions of tumor invasion. Diffuse WGA staining in tumor cell clusters is accompanied by marked cytoplasmic redistribution of E-cadherin, β-actin and syndecan-1 and by marked changes in cell morphology from polygonal to elongated.

WGA binds to glycoproteins in the cell membrane, and particularly to proteins involved in cell–cell adhesion: among the main surface proteins bound by WGA in macrophages and neutrophils is the cell–cell adhesion protein LFA-1 [8]. In endothelial cells, WGA interacts strongly with CD31, one of the main cell-cell adhesion proteins in the endothelium [9]. In the slime mold *Dictyostelium discoideum*, WGA binds to the cell–cell adhesion proteins responsible for the formation of multicellular aggregates [10]. Our results show that strong WGA staining at cell boundaries parallels membrane localization of E-cadherin, whereas loss of WGA staining at the cell surface is paralleled by loss of E-cadherin membrane localization. Taken together, these results suggest that a strong WGA signal at cell boundaries indicates an important presence of cell–cell adhesion complexes in these regions, and therefore high levels of tissue cohesion. Conversely, loss of WGA staining at cell boundaries, accompanied by diffuse cytoplasmic staining, could indicate diminished levels of cell–cell adhesion, and therefore decreased tissue cohesion. In support of this interpretation, mice deficient in Ephrin-B1, an important regulator of cell adhesion, show a marked shift from peripheral to cytoplasmic WGA staining pattern in conditions of tissue cohesion impairment [11]. Staining with biotinylated or horseradish peroxidase-conjugated WGA and visualization by bright-field microscopy of normal oral mucosa shows a similar pattern of strong staining at cell boundaries to the one reported here [15,16]. Interestingly, staining with biotinylated lectins in OSCC showed increased cytoplasmic staining and decreased membrane staining in invasive regions, in agreement with our results [17].

In regions of tumor invasion, we observed tumor cell clusters of varying sizes and patterns of WGA staining, which were readily discernible from the surrounding stroma. The presence of tumor buds, defined as individual cells or clusters of up to four cells, correlates with higher probability of metastasis in OSCC, and in many other cancers [18]. Scoring of tumor buds is done mainly by hematoxylin and eosin (H&E) staining. Alternatively, scoring can be done by cytokeratin immunohistochemistry, in situations when H&E staining does not allow to clearly distinguish buds from the surrounding stroma, such as when high levels of inflammation are present, or the tumor has a desmoplastic reaction [19]. Fluorescently labeled WGA staining is a simpler and faster procedure than cytokeratin immunohistochemistry, which warrants further investigation into its potential use for tumor bud scoring.

Cells in invasive tumor clusters shift E-cadherin localization from membranous to cytoplasmic and acquire a more elongated, spindle-like morphology [20]. These changes are indicative of a process of epithelial to mesenchymal transition (EMT), found in aggressive carcinomas such as OSCC [21], which in invasive tumor clusters has been proposed to lead to an intermediate epithelial/mesenchymal phenotype, associated with patterns of collective cell migration rather than generalized, individual tumoral cell migration and dissemination [22]. Consistent with these results, we have also observed remarkable E-cadherin cytoplasmic staining in cells from tumor clusters, and frequent examples of cells shifting from a polygonal to an elongated morphology, accompanied by a marked decrease of WGA staining at cell boundaries. We detected prominent cytoplasmic β-actin staining in cells from tumor clusters as well, associated with elongated cell morphologies, and a shift from cortical to cytoplasmic actin is another feature of EMT [23]. Loss of polygonal cell morphologies was also observed in tumor cells with strong cytoplasmic syndecan-1 expression, and cytoplasmic redistribution of syndecan has been linked with increased tumor invasiveness [24]. Our results showing co-expression of vimentin and pan-cytokeratin and co-expression of vimentin and Ki-67 in tumor clusters are also consistent with the acquisition of mesenchymal properties, while still retaining epithelial characteristics and the proliferative activity found in OSCC invasive regions with high numbers of tumor clusters [25].

Interestingly as well, we have observed tumoral cell clusters with well-defined WGA staining at cell boundaries (Figure 2D), whereas other clusters show a much more diffuse WGA staining pattern, marked by prominent cytoplasmic E-cadherin, β-actin and syndecan-1 staining (Figure 5). These results are consistent with variable cell cohesion in tumor clusters of OSCC, which has been reported in other cancers and might be linked to different stages in the process of EMT, which is also consistent with the observation of an invasion pattern spectrum in OSCC, with some tumor cells being more advanced in the transition towards a mesenchymal phenotype than others [20,26]. It is also important to consider that the tumor–stroma ratio is an important parameter with prognostic value in OSCC [27], and as WGA staining readily allows one to distinguish between tumor clusters and surrounding stroma, it could be applied to quantify this variable as well.

Therefore, our results analyzing WGA and structural proteins staining patterns suggest that changes in cell cohesion occur at different stages of OSCC progression—first in the basal layers of the oral mucosa and then in invasive regions, where tumoral cell clusters are found. Changes in cell cohesion at these different stages might underlie first the initial invasion into the stroma and then the further dissemination of tumor cells within the stroma.

## 5. Conclusions

In summary, the use of WGA staining, in combination with TSA staining of structural proteins and confocal microscopy, constitutes a valuable tool for the study of OSCC tissue architecture. Further studies will be needed to define possible uses of fluorescent WGA staining in combination with confocal microscopy in the histopathological assessment of OSCC.

## Figures and Tables

**Figure 1 cells-10-02466-f001:**
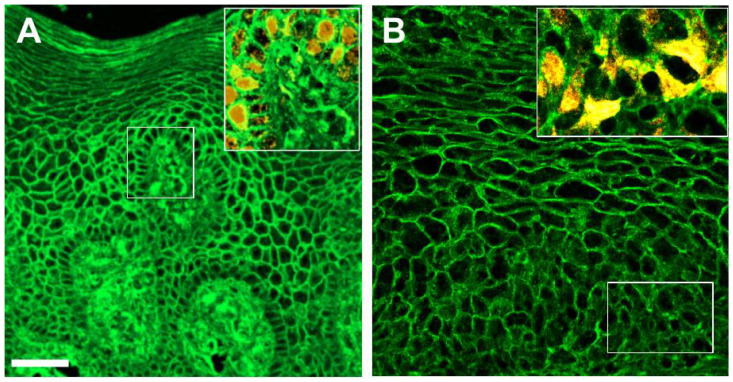
WGA staining pattern (green) in normal oral mucosa (**A**) and in OSCC mucosa (**B**), with a 20× objective. Insets show the regions marked with white rectangles observed with a 40× objective, displaying Ki-67 staining (yellow) in addition to WGA staining. Scale bar: 50 μm.

**Figure 2 cells-10-02466-f002:**
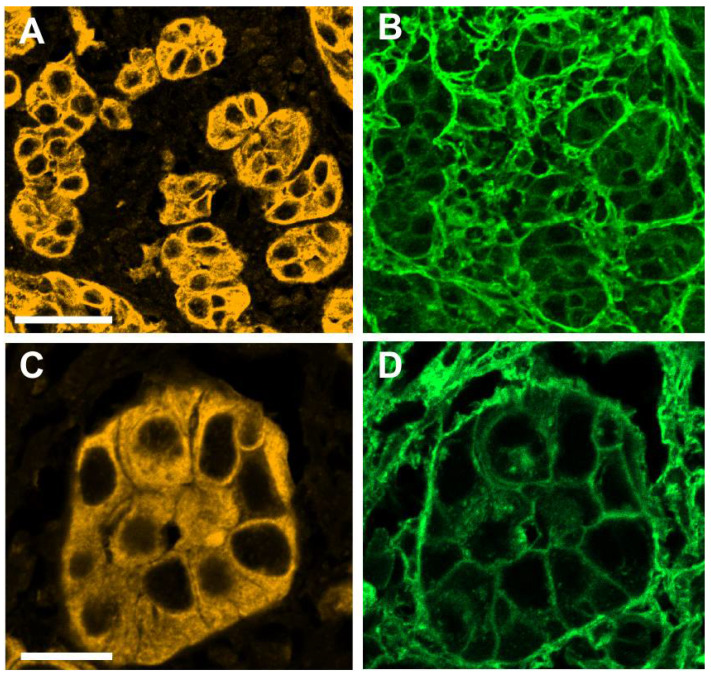
Pan-cytokeratin (yellow) and WGA staining (green) patterns in invasive regions of OSCC. (**A**,**B**) show the same field with pan-cytokeratin and WGA staining, respectively, with a 20× objective. (**C**,**D**) show the same field, with pan-cytokeratin and WGA staining, respectively, with a 40× objective. Scale bars: (**A**) 50 μm; (**B**) 20 μm.

**Figure 3 cells-10-02466-f003:**
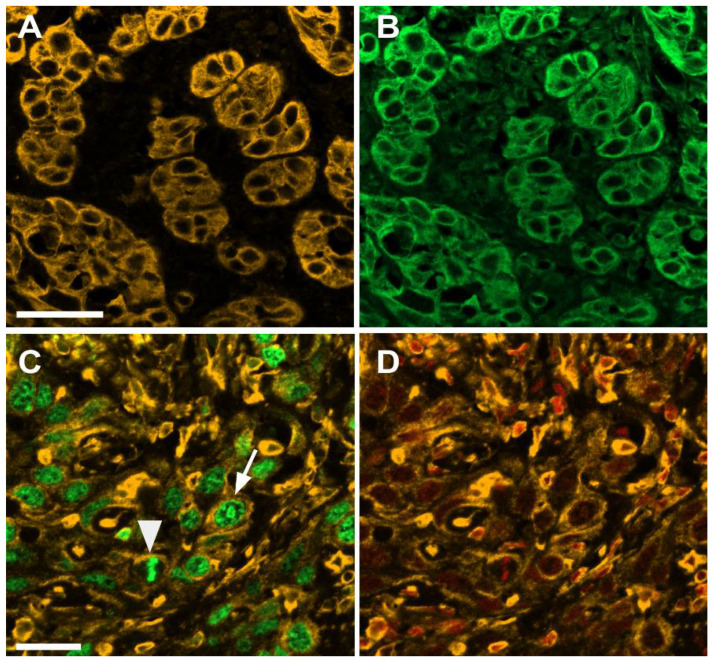
Pan-cytokeratin, vimentin and Ki-67 staining patterns in invasive regions of OSCC. (**A**,**B**) show the same field with pan-cytokeratin (yellow) and vimentin (green) staining, respectively, with a 20× objective. (**C**,**D**) show the same field with vimentin (yellow) and Ki-67 (green) or Far-Red Nuclear Stain (red) staining, respectively, with a 40× objective. The arrow in (**C**) shows a large cell with epithelial morphology positive for vimentin and Ki-67. The arrowhead in (**C**) shows a large cell positive for vimentin and Ki-67 in metaphase. Scale bars: (**A**) 50 μm; (**C**) 20 μm.

**Figure 4 cells-10-02466-f004:**
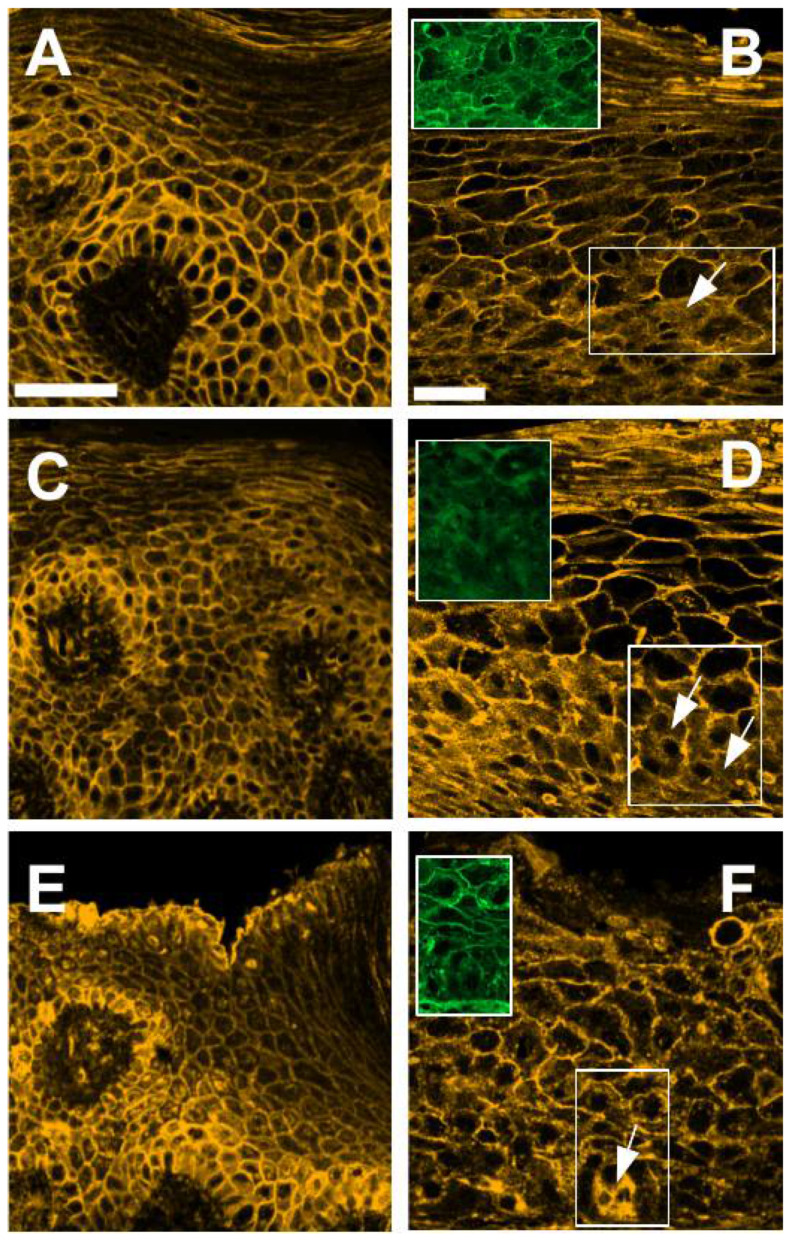
E-cadherin, β-actin and syndecan-1 staining patterns (yellow) in normal oral mucosa (**A**,**C**,**E**) and in OSCC mucosa (**B**,**D**,**F**), respectively, with a 40× objective. (**B**,**D**,**F**) were taken more zoomed in than (**A**,**C**,**E**). Insets in (**B**,**D**,**F**) show the regions marked with white rectangles in the basal layer of OSCC mucosa, displaying WGA (green) staining. Arrows show regions with marked cytoplasmic staining and diffuse cell borders. Scale bars: (**A**,**C**,**E**) 50 μm; (**B**,**D**,**F**) 50 μm.

**Figure 5 cells-10-02466-f005:**
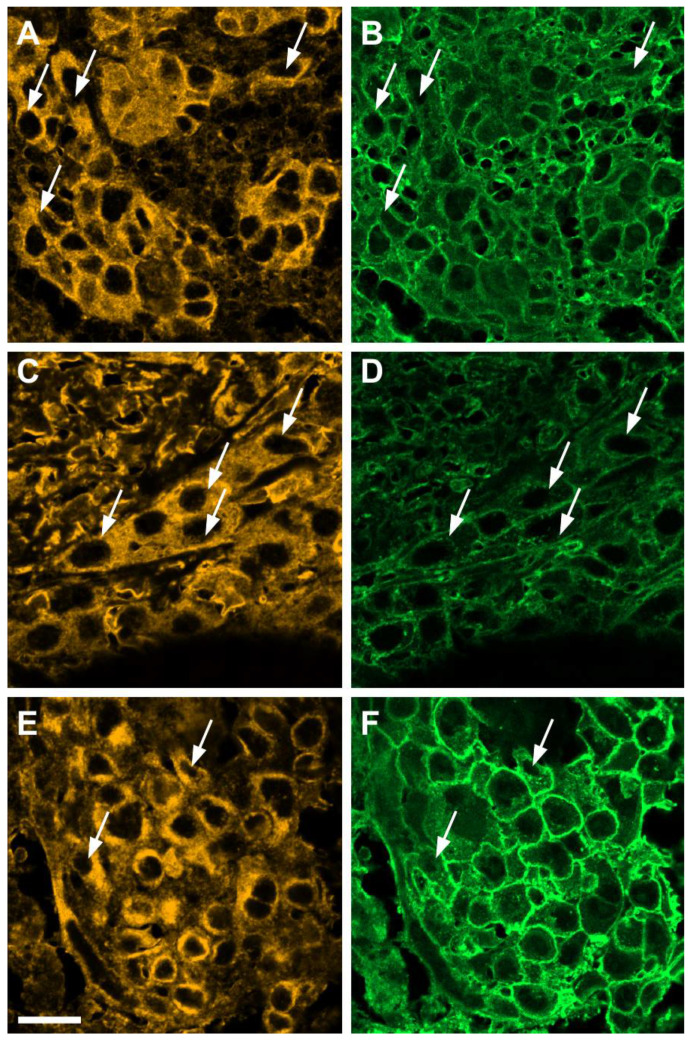
E-cadherin, β-actin and syndecan-1 staining patterns (yellow) in invasive regions of OSCC, with a 40× objective. (**A**,**B**): same field with E-cadherin and WGA (green) staining, respectively. (**C**,**D**): same field with β-actin and WGA staining, respectively. (**E**,**F**): same field with syndecan-1 and WGA staining, respectively. Arrows mark cells with highly elongated morphology and abundant cytoplasmic E-cadherin (**A**), β-actin (**B**) or syndecan-1 (**E**). Scale bar: 20 μm.

## Data Availability

Not applicable.

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
