# Peer review of "The Tissue Architecture of Oral Squamous Cell Carcinoma Visualized by Staining Patterns of Wheat Germ Agglutinin and Structural Proteins Using Confocal Microscopy"

_cells, 2021, doi:10.3390/cells10092466_

Round 1

Reviewer 1 Report

In this manuscript the authors investigated tissue architecture in oral cancer by using confocal microscopy to examine the staining pattern of wheat germ agglutinin (WGA),and of other structural proteins.

The results were interesting, suggesting that a strong WGA signal at cell boundaries indicates a strong cell-to-cell adhesion complexes, and therefore high levels of tissue cohesion.

Although the number of patients is small, this is an interesting study, conducted with a rigorous method. The text clear and easy to read and the Discussion is quite balanced. Overall, this study represents an excellent starting point for further studies.

The Authors reported the role of tumor buds and EMT process in oral cancer. As the importance of the topic, I suggest adding some considerations regarding the prognostic role of stromal component of oral squamous cell carcinoma, as recently reported in literature (for your convenience, DOI: 10.1111/his.14202).

There are some typos in the text: minor language corrections should be necessary to improve readability.

Author Response

We are grateful for the referees comments, which we have addressed in the new version of the manuscript, as described below:

"The Authors reported the role of tumor buds and EMT process in oral cancer. As the importance of the topic, I suggest adding some considerations regarding the prognostic role of stromal component of oral squamous cell carcinoma, as recently reported in literature (for your convenience, DOI: 10.1111/his.14202)."

R= The reference as well as considerations regarding the prognostic role of the tumor-stroma ratio in OSCC have been added to the discussion

"There are some typos in the text: minor language corrections should be necessary to improve readability."

R= We have corrected typos and replaced "tumour" with "tumor" throughout the text

Reviewer 2 Report

The authors could review spelling of some words in the text of manuscript  - example tomour into often used -  tumor

Author Response

We are grateful for the referees comments, which we have addressed in the new version of the manuscript, as described below:

"The authors could review spelling of some words in the text of manuscript - example tomour into often used - tumor"

R= We have corrected typos and replaced "tumour" with "tumor" throughout the text

Reviewer 3 Report

Tyramide signal amplification is typically used in connection with a staining protocol. e.g. Indirect immunofluorescence with tyramide signal amplification.

Describe the carbohydrate moiety that WGA binds to and why this is significant in these experiments.

Letters on images MUCH to big  and scale bar often too long.

MUST TELL THE READER WHICH COLOR IS WHICH STAIN - ESPECIALLY IN FIG 3.

Fig 1: why is WGA in cytoplasm?

Fig 2: Suggest add a normal oral mucosa example - more helpful than a 20X and 40X comparison of the same field of view.

Figs. Description of magnification is incorrect in figure legends and correct in Material and Methods. I would also suggest adding the identification of the objectives in the M and M. In the figure legends, instead of ", at 20X magnification" say with a 20X objective.

Fig 4 - unclear what we are to see - arrows would be helpful. Include that this is in the basal layer and in Fig 5 legend include that is in regions of tumor invasion. I can see cells in A, C and E that have "filled" cytoplasm. Seems like the normal and the OSCC mages are taken at different magnification or with different zooms?.

Fig 5: Add one set of normal mucosa for comparison to this set.

The image, as presented in this format, are not convincing and don't support the authors conclusions. The data is probably there but not as currently presented. Also, don't need to do confocal microscopy with 5 micron sections- this is doable with a conventional scope with appropriate filter sets.

Author Response

We are grateful for the referees comments, which we have addressed in the new version of the manuscript, as described below:

"Tyramide signal amplification is typically used in connection with a staining protocol. e.g. Indirect immunofluorescence with tyramide signal amplification"

R= We have corrected this in the text, using for example "immunohistochemistry with tyramide signal amplification", or "immunohistochemistry accompanied by tyramide signal amplification"

"Describe the carbohydrate moiety that WGA binds to and why this is significant in these experiments."

R= WGA binds to proteins with varying carbohydrate moieties containing N-acetylglucosamine and sialic acid. This point has been added in the introduction

"Letters on images MUCH to big  and scale bar often too long"

R= This has been corrected in all the figures

"MUST TELL THE READER WHICH COLOR IS WHICH STAIN - ESPECIALLY IN FIG 3. "

R= This information has been added in all the figure legends

"Fig 1: why is WGA in cytoplasm?"

R= As stated in the manufacturer protocol for fluorescently labeled WGA, "In permeabilized cells, WGA will label glycoproteins in the plasma membrane and intracellular compartments such as Golgi structures." (https://biotium.com/wp-content/uploads/2016/12/PI-WGA-Conjugates.pdf). Therefore, WGA staining in the cytoplasm would be expected to occur at least to some degree.

"Fig 2: Suggest add a normal oral mucosa example - more helpful than a 20X and 40X comparison of the same field of view."

R= Figure 2 shows pan-cytokeratin and WGA staining of invasive OSCC regions. We haven't been able to perform pan-cytokeratin staining in normal oral mucosa in the time frame required for this response due to an outbreak of Covid-19 in our institution. However, we would expect normal oral mucosa to show positive pan-cytokerain staining, as demonstrated for instance by Hsueh et al (DOI: 10.1038/srep36266).

"Figs. Description of magnification is incorrect in figure legends and correct in Material and Methods. I would also suggest adding the identification of the objectives in the M and M. In the figure legends, instead of ", at 20X magnification" say with a 20X objective."

R= This has been corrected in all the figure legends

"Fig 4 - unclear what we are to see - arrows would be helpful. Include that this is in the basal layer and in Fig 5 legend include that is in regions of tumor invasion. I can see cells in A, C and E that have "filled" cytoplasm. Seems like the normal and the OSCC mages are taken at different magnification or with different zooms?."

R= We have included arrows in figure 4, an indication in the legend that it is the basal layer in OSCC mucosa, as well as indications in the results sections that in OSCC, strong cytoplasmic signals for E-Cadherin, beta-actin and syndecan-1 are accompanied by blurring of cell boundaries, a pattern not seen in normal oral mucosa. We have also added a mention in the legend of figure 4 regarding the larger zoom used for OSCC images, as well as the appropriate scale bar for these images. In figure 5, the legend states that images correspond to regions of tumor invasion.

"Fig 5: Add one set of normal mucosa for comparison to this set".

R= We have added Supplementary figure 1, showing patterns of E-cadherin, beta-actin and syndecan-1 in normal oral mucosa at high magnification.